# Intracranial Meningioma in Elderly Patients. Retrospective Multicentric Risk and Surgical Factors Study of Morbidity and Mortality

**DOI:** 10.3390/diagnostics12020351

**Published:** 2022-01-29

**Authors:** Daniele Armocida, Umberto Aldo Arcidiacono, Mauro Palmieri, Alessandro Pesce, Fabio Cofano, Veronica Picotti, Maurizio Salvati, Giancarlo D’Andrea, Diego Garbossa, Antonio Santoro, Alessandro Frati

**Affiliations:** 1Neurosurgery Division, Human Neurosciences Department, “Sapienza” University, 00185 Rome, Italy; arcidiacono.md@gmail.com (U.A.A.); mauro.palmieri@uniroma1.it (M.P.); antonio.santoro@uniroma1.it (A.S.); alex.frati@gmail.com (A.F.); 2Santa Maria Goretti Hospital, 04100 Latina, Italy; ale_pesce83@yahoo.it; 3Neurosurgery Unit, Department of Neuroscience “Rita Levi Montalcini”, University of Turin, 10124 Turin, Italy; fabio.cofano@gmail.com (F.C.); diego.garbossa@unito.it (D.G.); 4Neurosurgery Department, Fabrizio Spaziani Hospital, 03100 Frosinone, Italy; veronica@picotti.com (V.P.); gdandrea2002@yahoo.it (G.D.); 5Policlinico Tor Vergata, University Tor Vergata of Rome, 00133 Rome, Italy; salvati.maurizio@libero.it; 6IRCCS “Neuromed”, 86077 Pozzilli, Italy

**Keywords:** meningioma, elderly patient, neurosurgery, brain tumor, risk factors

## Abstract

With the increasing life expectancy, a large number of intracranial meningiomas (IM) have been identified in elderly patients. There is no general consensus regarding the management for IMs nor studies regarding the outcome of older patients undergoing meningioma surgery. We aimed to determine whether preoperative variables and postoperative clinical outcomes differ between age groups after meningioma surgery. We analyzed data from all patients who had undergone IM surgery from our departments. The final cohort consisted of 340 patients affected by IM with ASA class I-II: 188 in the young group (<65) and 152 in the elderly. The two subgroups did not present significant differences concerning biological characteristics of tumor, localization, diameters, lesion and edema volumes and surgical radicality. Despite these comparable data, elderly presented with a significantly lower Karnofsky Performance status value on admission and remained consistently lower during the follow-up. We establish instead that there is no intrinsic correlation to the presence of IM and no significant increased risk of complications or recurrence in elderly patients, but rather only an increased risk of reduced performance status with mortality related to the comorbidity of the patient, primarily cardiovascular disease, and an intrinsic frailty of the aged population.

## 1. Introduction

Meningiomas are the most common benign intracranial tumors, which may occur at any age, but in actuality, they present an incidence peak between the sixth and seventh decades of life [1,2], and the incidence rate increases progressively with increasing age [3]. With the availability of advanced diagnostic imaging modalities and an increasing human life expectancy, a large number of intracranial meningiomas (IM) have been identified in elderly patients [4,5]. Elderly patients were described as those aged over 65 years old [6,7,8,9,10].

Although the principle of surgical treatment is to achieve as much tumor removal as possible with the least morbidity and mortality, published reports in the existing literature have revealed a wide range of mortality (0–45%) and complication (10–39%) rates in elderly patients with surgical resection of IMs [6,7,8,9,11,12,13,14].

There is still no general consensus regarding surgical management for IMs in elderly patients, and the predictive factors of elderly patients with surgically resected IMs are not well established [13]. This raises the question of whether the surgical procedure, as a first treatment option, is as good for older patients as it is for younger adult patients, and whether the postoperative clinical outcome in these patients, in terms of complications, surgical morbidity, and mortality, is comparable to that of younger patients [15].

Some authors have found that despite the differences in ASA classification and preoperative Karnofsky performance status (KPS), there was no significant difference in the mortality, complication rate, and long-term outcome between younger and older adults after surgery for IM [15]. Upon analyzing the literature to define the morbidity and mortality of elderly patients who underwent surgery for IM, several authors identify in their analysis all clinical, radiological, and biological variables, focusing more on patients’ comorbidity favoring their results of surgery.

In this study, we aimed to understand which risk factors are associated with post-surgical outcomes and how these risk factors affected post-surgical outcomes in elderly patients with IM. We compared data of younger and elderly patients operated on for IM on the grounds of clinical, radiological, and surgical features of a large multicentric series from two departments of Neurosurgery of the University Sapienza of Rome, Italy.

## 2. Materials and Methods

The aim of this study is to determine whether preoperative variables and postoperative clinical outcome differ between age groups after meningioma surgery.

### 2.1. Participants and Eligibility

We analyzed the data from all patients who had undergone IM surgery from 2016 to 2020 in our neurosurgery department at the Sapienza Neurosurgery department of Rome (Italy) and the Neurosurgery department of Hospital Spaziani of Frosinone (Italy).

We collected a total of 472 patients suffering from IM. Histological diagnoses were performed according to the updated version of the 2021 WHO guidelines [16]. Patients were subject to the following inclusion and exclusion criteria:

Patients were included in the study if their pre- and post- operative magnetic resonance imaging (MRI) was either performed at our institution or available on the picture archiving and communication system (PACS) for review.

If, in the postoperative period, they could undergo a standard clinical and radiological follow-up starting from the 30th day after surgery.

The estimated target of the surgical procedure was the total or subtotal resection of the lesions (no biopsies were included).

Patients with severe comorbidity (with the valuation of ASA class III-V) that may compromise the evaluation in follow-up (intractable oncological, metabolic, or cardiovascular diseases) were excluded.

Patients were excluded for incomplete or wrong data on clinical, radiological and surgical records and/or lost to follow-up.

All the patients who met the aforementioned inclusion criteria were investigated regarding whether increasing age is indicative for different overall survival (OS), grading, location, clinical onset, performance status, immunohistochemical and radiological characteristics, and clinical/neurological outcome. Patients were divided into two groups for analysis: Group A patients <65 years and Group B patients >65 years. Outcome analyses were performed by additionally comparing the division of the cohort to 4 groups (<45 years, 45–65 years, 65–75 years, >75 years).

The ASA classification was used as a system for assessing the general health condition of patients before surgery. No patient from the latter group underwent meningioma surgery [11].

For all the included patients, we first recorded their age, sex, time of hospitalization, time of follow-up, clinical onset, presence of smoke habits, cardiological diseases, and performance status (measured using KPS at the moment of radiological diagnosis).

Regarding the clinical onset, we considered focal neurological deficits, the focal disorders of body motility and sensitivity, sphincter disorders, disorders involving cranial nerves including visual disturbances, the presence of dizziness, alteration of mental status and memory loss, the presence of intractable headache, seizure, and the incidental diagnosis.

Upon radiological evaluation, we recorded the location of the lesion, tumor major diameter (measured in cm), tumor volumes (measured in cm^3^), Edema volume (measured in cm^3^ and before anti-edemigen therapy), the presence of multiple IMs and or meningiomatosis, and the involvement of the subtentorial compartment.

On the grounds of the final histological diagnosis, we recorded the WHO grading with subtypes, the mitotic index measured using the count of mitosis on 10 high-power fields (HPFs), and immunohistochemistry with ki67 and Progesteron (PR) expression routinely performed in the Department of Neuropathology of our Hospital; Ki67 was applied to frozen sections of fresh tissue using a standard immunoperoxidase technique.

OS was recorded in months, and was measured from the date of diagnosis to the date of death or the date of last contact if alive. Clinical information was obtained by the digital database of our Institution, whereas OS data were obtained by telephone interviews. After the surgical procedure, we recorded the status of performance (using KPS) for each patient at 1 month, 6 months, and at the last clinical evaluation. A special focus was on the KPS results: Such a parameter was considered as it has previously been observed as predictive and associated with survival (methodology described for other studies reported [17,18]). We evaluated the presence of complications, recurrence, and eventual second treatment, recording the biological switch.

All the patients underwent a preoperative brain MRI scan that included a high-field 3 Tesla volumetric study with the following sequences: T2w, FLAIR, and isotropic volumetric T1-weighted magnetization-prepared rapid acquisition gradient echo (MPRAGE) before and after intravenous administration of paramagnetic contrast agent. The volumes of the contrast-enhancing lesion and edema were calculated drawing a region of interest (ROI) in a volumetric-enhancing post-contrast study weighted in T1 (a multi-voxel study) and T2, conforming to the margins of the contrast-enhancing lesion with the software Horos (an ROI and volume calculation sample is reported in Figure 1) [19]. On the first postoperative day, the patients underwent a CT head scan to retrieve early complications [20] and a volumetric Brain MRI scan to evaluate the extent of resection (EOR) and measure the Simpson grade.

### 2.2. Statistical Methods and Ethics

The sample was analyzed with SPSS version 18. Comparisons between nominal variables have been made with the Chi2 test. EOR (measured by the Simpson Grade) means were compared with One-Way and Multivariate ANOVA analysis along with Contrast analysis and Post-Hoc Tests. Continuous variables’ correlations have been investigated with Pearson’s Bivariate correlation. The threshold of statistical significance was considered *p* < 0.05.

Informed consent was approved by the Institutional Review Board of our Institution. Before the surgical procedure, all patients gave informed written explicit consent after the appropriate information was provided. The data reported in the study have been completely anonymized.

For statistical analyses, data collection, and the analysis of results, we received support from the Neurosurgical department of Turin, Italy, directed by Prof. D. Garbossa. No treatment randomization was performed due to the retrospective nature. This study is perfectly consistent with the Helsinki declaration of Ethical principles for medical research on humans.

## 3. Results

### 3.1. Descriptive Data

The final cohort consisted of 340 patients (102 males and 238 females—70% of the population), and the average age was 60.38 ± 13.56 years (range 20–90).

In the final division in the main subgroup A, the number of meningiomas in the young group <65 were 188, and in subgroup B, meningiomas in the elderly totaled 152. All the relevant details with analysis results are included in Table 1.

The two subgroups did not present marked differences from the time of hospitalization and follow-up (0.19 and 0.639, respectively). Smoking habits were revealed at the time of radiological diagnosis in 52 patients (34%) and 46 patients (40%), respectively, without significant differences (*p* = 0.199). Cardiological diseases such as hypertension, a history of ischemia, or arrhythmia were identified in 29 patients (21%) and 79 patients (68.1%), respectively, with a significant difference in the group of elderly patients (*p* < 0.01).

### 3.2. Histochemical Comparison Analisys between the Two Groups

From the histochemical point of view, concerning the WHO classification, the two subgroups did not present statistically significant differences in terms of grading (*p* > 1, Figure 2), the expression of ki67 (*p* = 0.847, Figure 3A), or the mitotic index measured on 10 HPF (*p* = 0.372, Figure 3B), although a progressive increase in these parameters among the age groups was observed.

The most frequent histological subtype in both groups was the grade I meningothelial form.

There is a statically significant difference in the incidence of IM in the female sex (*p* = 0.09) in the group of young patients (Group A) that also correlates with a greater presence of progesterone positivity (*p* = 0.041 one-sided) indicative of a likely greater hormone dependence of the tumor compared to elderly patients (Group B).

### 3.3. Radiological Comparison Analisys between the Two Groups

From the radiological point of view, upon analyzing the data of the MRI image performed at the time of diagnosis and without having started any drug therapy, the two subgroups showed no significant differences in localization (*p* = 0.856, Figure 4), the presence of meningiomatosis or multiple lesions (*p* = 0.434), and in the mean values of diameter (*p* = 0.693), lesion volume (*p* = 0.893), and corresponding edema present at the measurements performed in T2-FLAIR (*p* = 0.902). No radiological pre-operative differences were observed in the division of the four sub-groups.

### 3.4. Outcome Data and Main Results

Both a two-group stratification of <65 and >65 years and a division into the four age groups were considered for the outcome study in relation to the radiological findings (data resumed in Table 2).

From the point of view of the surgical radicality achieved with the surgery, there are no significant differences in obtaining a Simpson type I grade assessed at the first postoperative MRI imaging between the two groups (*p* = 0.138 chi-square analysis) or the Simpson grading differences between groups (ANOVA study. *p* = 0.611). This finding also logically correlates with an insignificant difference in the recurrence rate (*p* = 0.598), and clinically, with the shift of the lesion to a higher grade and/or malignancy (*p* = 0.599).

Interestingly, despite a comparable class of ASA between the two groups (having selected only grades I and II for analysis), patients in group B presented to the intervention with a significantly lower KPS value (*p* < 0.01). Recovery of neurological function and performance grade after the surgical procedure remained consistently lower in the elderly group both in the postoperative phase and in the last evaluation at follow-up (both *p* < 0.01). The same type of statistical difference was obtained if cardiac patients in each group were excluded from the analysis (*p* < 0.001). These data are also confirmed when the analysis was performed considering the four age groups (Figure 5). These data do not correlate with the significantly increased incidence of postoperative operative complications (*p* = 0.364, Figure 6) or with the higher rate of ischemia (*p* = 1), hemorrhage (*p* = 0.41), infections (*p* = 0.314), and seizures (*p* = 0.217) in the first 30 days after surgery (Figure 7).

Measuring and considering all early and late complications in their totality during the entire follow-up (*p* = 0.039 one-sided) and considering the overall mortality (*p* < 0.01), there are statistically significant differences between patients <65 years and >65 years. Through the ANOVA study, it was found that this difference is maintained even when analyzing complication and mortality rates over the four age subdivisions (*p* < 0.01), thus showing no differences in the groups between 65–75 and >75 years (Figure 8).

Finally, in the multivariate analysis, although a correlation is seen between an increasing ki67 value and the mitotic index with increasing age, these parameters are not related to edema volume, the recurrence rate, the Simpson grade, or survival.

## 4. Discussion

Therapeutical management of meningiomas in elderly patients remains difficult due to controversial outcomes of surgery in this age group, and there are still conflicting reports in the existing literature. Some authors [3,8,15,21,22] suggest that meningioma surgery is well tolerated in elderly patients, and age alone is not a contraindication for surgery [8] and outcomes are therefore largely influenced by the characteristics of the tumor itself and the ASA class [3]. In the most complete meta-analysis published in 2021 by Rafiq et al. [23], it was defined that the elderly group had a high-risk ratio of being in ASA classes 3–5 in comparison to the young group, and the higher mortality rate observed in the elderly is likely secondary to the higher incidence of comorbidities.

In our study, which compares two groups with a perfectly comparable ASA score, it is demonstrated that in the elderly subjects, there is a greater risk of complications and greater difficulty in post-operative recovery.

In our opinion, the reason surgery in the elderly is still to be considered higher risk lies in the broadest possible consideration of all the other possible factors that influence the outcome. Different works [24,25,26,27] have previously attempted to identify and define prognostic factors for surgical outcomes in elderly patients, but information about patient selection for surgical treatment is limited by different study designs and indications for surgery across neurosurgical centers, showing conflicting, and often incomplete, data.

Several other factors were predictive of outcome, including patient factors, male sex, neurological conditions, general health indicators such as KPS score, and tumor parameters [12], such as the size and location, presence, and severity of edema radiological features, extension, grading [4,13,28,29,30], and operative approach [31].

While operative complications (intracranial hematoma, hydrocephalus, and focal neurologic deterioration) are strongly associated with surgical strategy and skill, non-operative complications, such as infection, atelectasis, and deep venous thrombosis, are highly associated with patients’ general condition and underlying diseases [32], correlated with age.

We confirm with our study that patients defined as elderly (with an age above 65 years) have a higher rate of late non-operative complications, higher mortality, and this independently concerns the comorbidity present, on the grounds of the same low ASA class, without a further distinction of age (elderly and ultra-elderly >75 years) [6,7,8,9], and, in contrast to the literature [26,33] irrespective of the biological behavior of the tumor.

It is interesting to note that in young subjects, especially in the female sex, the hormone dependence is an important factor in the presence and growth of meningioma compared to the elderly population. From the biological point of view, the higher rate of women in adulthood with higher progesterone expression explains the higher hormone dependence of IMs in young people without a significant impact on the outcome (Figure 9). Furthermore, we did not identify significant differences in the grading of lesions with a rate of high-grade lesions comparable between two groups.

The results confirm that older people might experience greater difficulty than younger patients, from mild to severe postoperative impairments of performance status, after tumor resection. We advocate a well-noted increased vulnerability to the stressors resulting from the poor general health and physiological reserve capacity of the elderly [15,34], and cardiological comorbidity conditions might result in a difference in the outcome [6,7].

In the present study, we demonstrated that IMs in elderly patients did not exhibit more frequent locations, a significantly larger tumor size, or a higher edema volume when compared with younger meningioma patients [28], although the performance statuses measured are significantly lower.

We confirm that the outcome of intracranial meningioma resection in the aged population is favorable when patients have no neurological deficits and a preoperative KPS score >70 [35], but when a decision regarding surgery is made for patients with a low preoperative KPS score or neurological deficits, we should consider additional factors, such as the preoperative general health condition, tumor characteristics, and patient perspectives. Although the use of new scoring systems may provide useful information for determining the optimal treatment for IMs in the elderly [13], in actual practice, the difficulties encountered during surgery for IMs [29], the technology available in the operating room, and the surgeon’s experience are more likely to affect the surgical outcome [11,13,28,29,36,37]. Our study showed factors that were related to surgical safety had a significant impact on the outcome, confirming that [13,32] factors associated with tumor features such as pathology and Simpson grades had less impact in terms of postoperative quality of life (QoL). So, currently, the best approach to select any treatment for a patient is a careful judgment of his general medical and neurological condition.

### Strengths, Limitations and Further Studies

Our study has some limitations. The small number of cases and lack of prospective studies limit the validity of the results. A comparative analysis between the surgically and conservatively or radio-surgically managed patients within each age group may contribute to a better understanding of mortality attributable directly to surgery.

The strengths of this study lie in the clinical setting, design, and follow-up. This is the only study comparing clinical, radiological, and surgical parameters in their entirety in a clinical outcome study between different age groups. The data were restricted to two hospitals only, belonging to the same University, thereby avoiding the selection bias inherently present in large multicenter studies. As the study includes all craniotomies performed for histologically verifiable IMs, there is no selection bias. All surgeries were performed within the same time period, thereby avoiding any time bias due to improvements or changes in neurosurgical care during the study period. Excluding patients with severe co-morbidities (with ASA III-V class assessment) could eventually bias the results, but it was necessary to clarify the variables that led to the outcome linked to surgery for IM.

## 5. Conclusions

With the increasing age of the population, the incidence of IMs in elderly people will continue to increase in the future. Although it is well established that IMs should be treated surgically when symptomatic [8], more postoperative complications are reported in patients older than 65 years, leading to an increased risk of mortality in these patients. For symptomatic IMs that are growing, radical surgical resection has been recommended as the initial treatment [3], but for this subgroup of patients, the general health condition, morbidities, and mortality rates in these age groups and the presumed postoperative outcomes regarding QoL should be carefully reviewed before determining the most appropriate treatment [38].

In our study, we establish instead that there is no intrinsic correlation (biology, volume, edema, tumor site) to the presence of IM and no significant increased risk of recurrence in elderly patients, but rather only an increased risk of reduced performance status with mortality related to the comorbidity of the patient, primarily cardiovascular disease, and the intrinsic frailty of the aged population. Clinical and neurological evaluation and operative risk assessment continue to play an important role in the decision-making process for surgery.

## Figures and Tables

**Figure 1 diagnostics-12-00351-f001:**
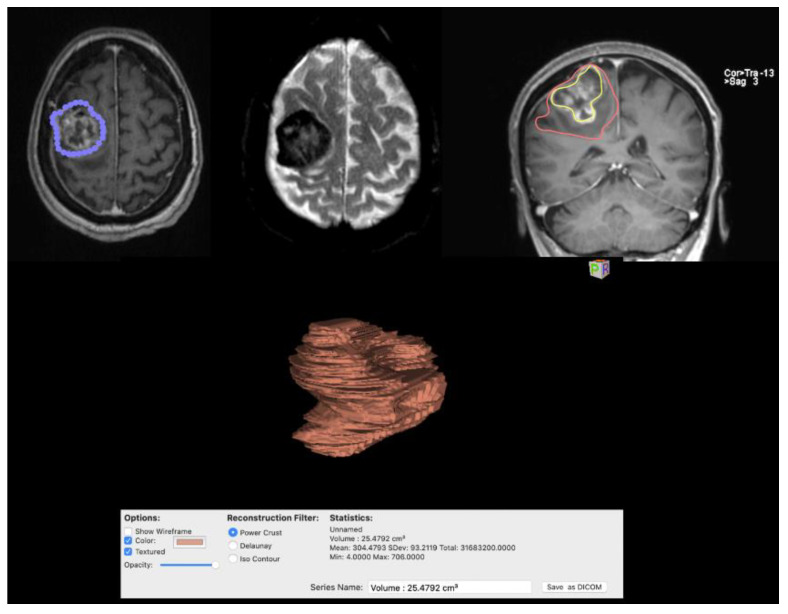
Images show calculation of edema volumes and contrast-capturing lesion with 3D reconstruction of the tumor obtained with Osirix software.

**Figure 2 diagnostics-12-00351-f002:**
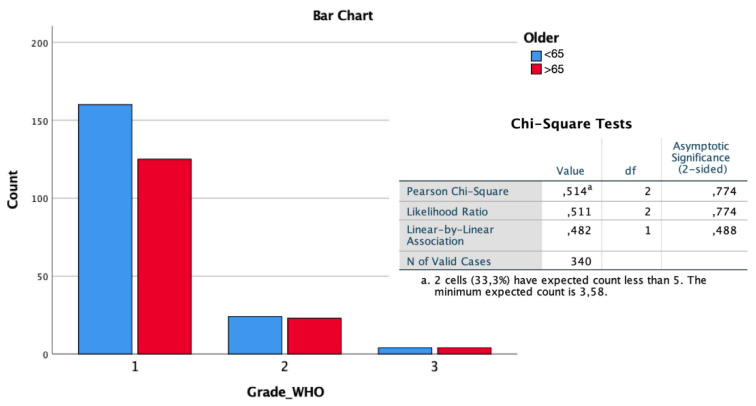
The distribution of the WHO grading found at histological analysis between young and elderly patients does not present statistically significant differences in the chi-square test.

**Figure 3 diagnostics-12-00351-f003:**
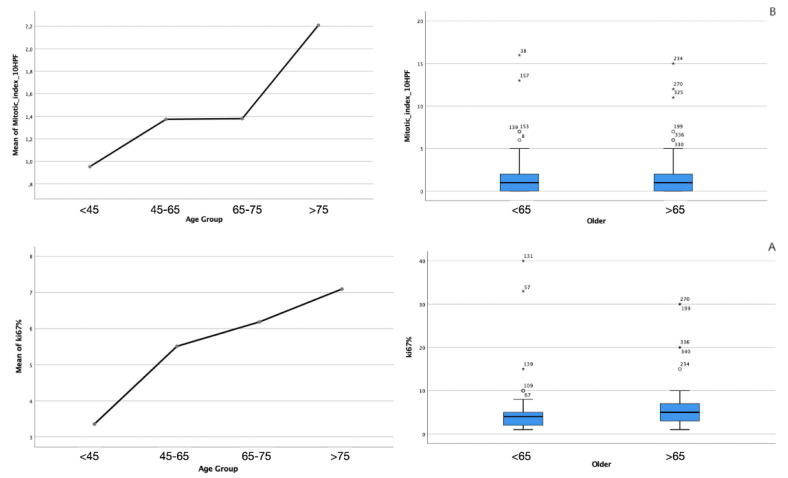
(**A**) ANOVA study performed analyzing ki-67 values and (**B**) observation of increasing number of mitoses per 10 HPF shows a progressive increase with increasing age. This finding does not correlate with grading or radiological features.

**Figure 4 diagnostics-12-00351-f004:**
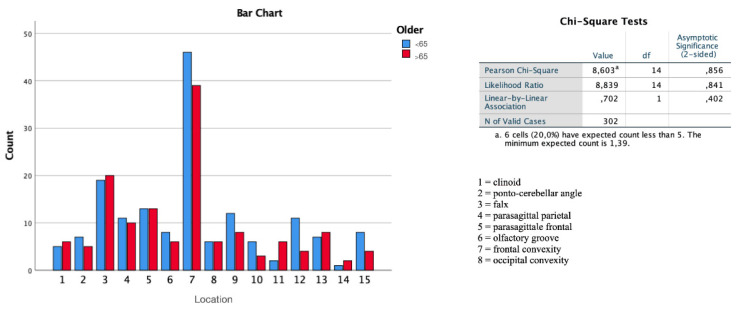
There is no more frequent intracranial localization in elderly patients than in young patients, upon analyzing individual cases with ANOVA study.

**Figure 5 diagnostics-12-00351-f005:**
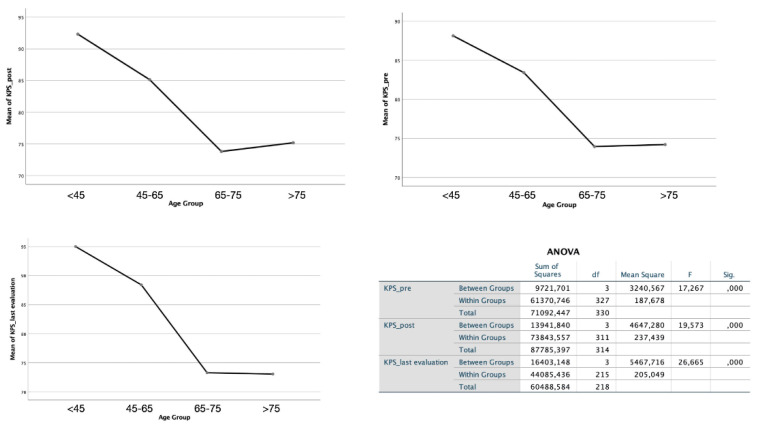
ANOVA study performed on the four age groups shows a significantly lower KPS value in patients over 65 years old. Further, recovery of neurological function and performance grade after the surgical procedure remained consistently lower in the elderly group both in the postoperative phase and in the last evaluation at follow-up (both *p* < 0.01).

**Figure 6 diagnostics-12-00351-f006:**
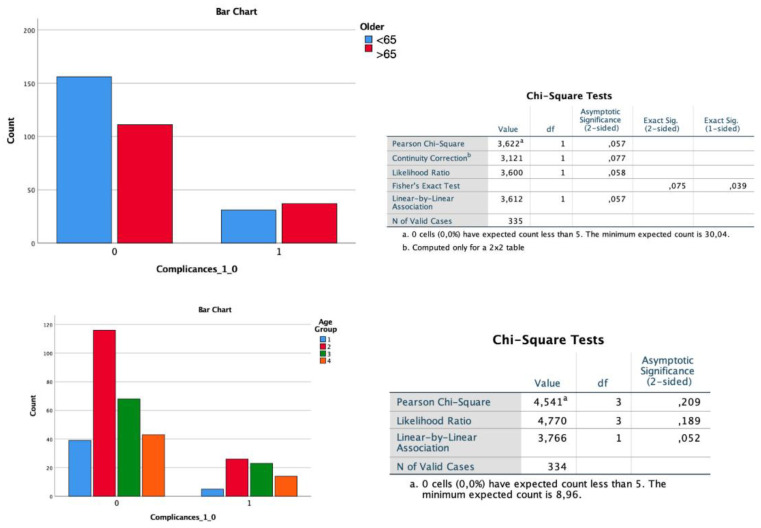
Analysis of complications with chi-square testing does not demonstrate the presence of an increased number of complications in the post-operative phase of young patients compared with the elderly.

**Figure 7 diagnostics-12-00351-f007:**
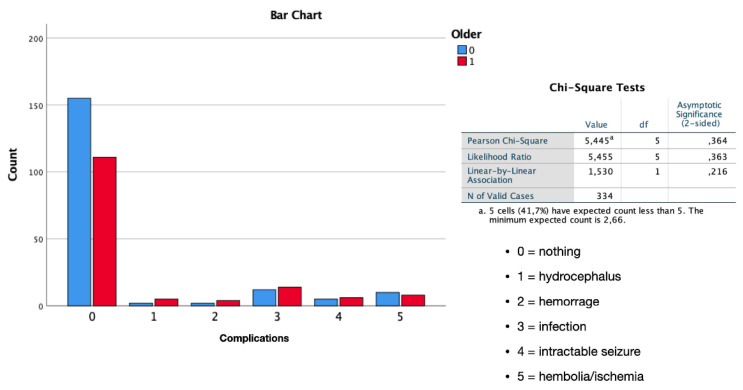
In the analysis of complications performed with ANOVA study, there is no one type of complication more frequent than another between the two groups.

**Figure 8 diagnostics-12-00351-f008:**
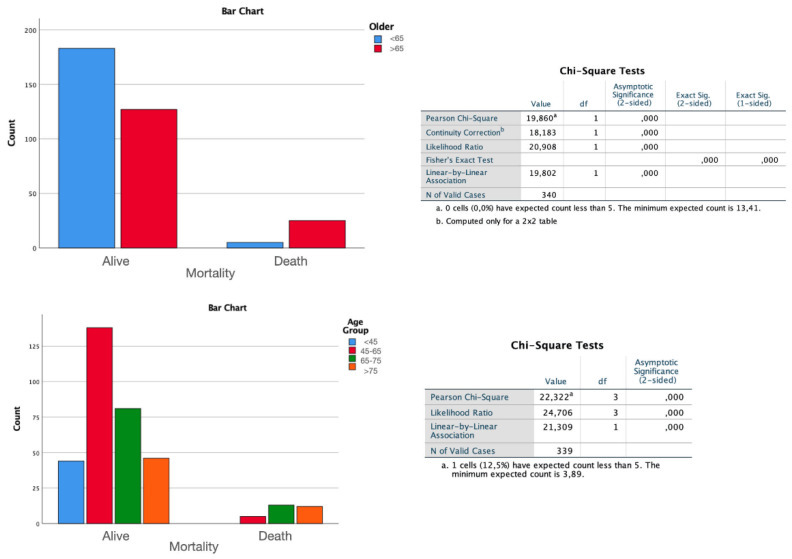
Bar chart of the ANOVA study shows that the difference in mortality at last evaluation is maintained even when analyzing rates over the four age subdivisions (*p* < 0.01), thus showing no differences in the groups between 65–75 and >75 years.

**Figure 9 diagnostics-12-00351-f009:**
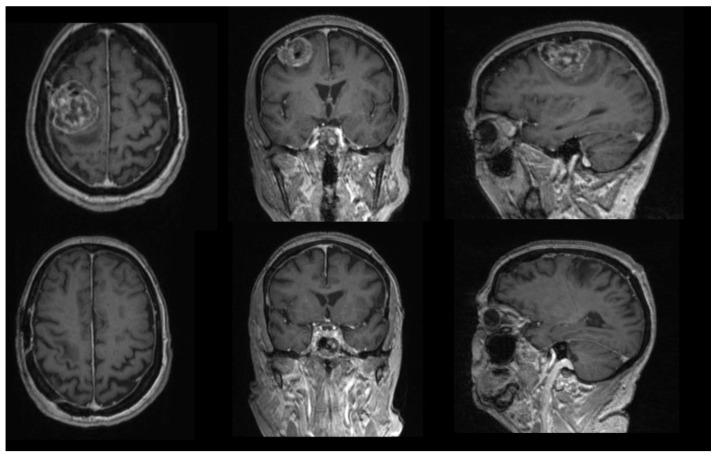
Images show the presence of a surgically treated right frontal parasagittal meningioma in an 80-year-old woman. Postoperative follow-up documents include a complete Simpson grade I resection; the procedure had no postoperative complications. Histologic examination confirmed the presence of grade I meningioma with progesterone negativity. The patient started experiencing seizures after 3 months. Sixteen months after the procedure, the patient died of cardiological and infectious problems.

**Table 1 diagnostics-12-00351-t001:** Patients populations with clinical, radiological, biological, and outcome analyses performed.

	Populations (340)	Group <65 (188)	Group Elderly (152)	*p*-Value
	Sex	F = 142	F = 96	0.09
Clinical	Hospitalization	15.79	20.26	0.19
	Follow-up	46.76 DS = 14.83	48.89 SD = 14.77	0.639
	Cardiopathy	29/140 = 20.71%	79/116 = 68.1%	<0.01
	Smoke habit	52/152	46/115	0.199
	Localization			0.856
Radiological	Multiple	8/188	5/145	0.434
	Diameter means	4.42	4.34	0.693
	Volume means	36.55	35.91	0.893
	Edema V means	27.99	28.78	0.902
	Ki67 means	5.03 SD= 6.34	6.58 SD = 6.43	0.847
Biological	Mitotic index 10 HPF	1.26 SD = 2	1.67 SD = 2.12	0.372
	PR +	27/175 = 15.42%	11/133 = 8.3%	0.079–0.041
	High grading (III-IV)	4/188 = 2.13%	4/152 = 2.63%	1
	Recurrence	10.5%	12.77%	0.598
	Simpson Grading			0.611
Outcome	Simpson Grade I			0.138
	Progression Malignant	3/188	2/145	0.599
	KPS pre	11	42	<0.01
	KPS post	14	41	<0.01
	KPS last	7	24	<0.01
	Complications post-operative			0.217
	Ischemia post	10/186 = 5.4%	8/148 = 5.4%	1
	Infections post	12/186 = 6.45%	14/148 = 9.45%	0.314
	Hemorrhage post	2/186 = 1.1%	4/148 = 2.7%	0.41
	Seizure post	22/186	19/145	0.364
	Complications Rate	31/187 = 16.6%	37/148 = 25%	0.075–0.039
	Mortality	5/188 = 2.6%	25/152 = 16.44%	<0.01

**Table 2 diagnostics-12-00351-t002:** Principal analysis of radiological and outcome data performed on four age groups.

Populations	Group 1 (<45) 44 pts	Group 2 (45–65) 143 pts	Group 3 (65–75) 94 pts	Group 4 (>75) 58 pts	*p*-Value
Diameter mean	4.39 SD = 1.75	4.43 SD = 1.88	4.31 SD = 1.73	4.38 SD = 1.44	0.971
Volume mean	35.70 SD = 29.58	36.80 SD = 41.22	37.87 SD = 37.46	32.96 SD = 28	0.919
Edema vol mean	23.63 SD = 35.78	29.25 SD = 47.97	24.75 SD = 43.85	35.23 SD = 53.94	0.701
Complicances					0.209
Mortality rate	0	5 = 3.5%	13 = 13.8%	12 = 20.7%	<0.01

## Data Availability

The dataset generated and analyses during the current study are not publicly available and are not retrieved for National databases, because they are the result of an institutional internal research of all treated cases of IM in our Hospital (Policlinico Umberto I of Rome and Spaziani Hospital of Frosinone). The original dataset is available from the corresponding author upon reasonable request.

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
