# Peer review of "Intracranial Meningioma in Elderly Patients. Retrospective Multicentric Risk and Surgical Factors Study of Morbidity and Mortality"

_diagnostics, 2022, doi:10.3390/diagnostics12020351_

Round 1
Reviewer 1 Report
This manuscript presents a retrospective study on intracranial meningioma patients who had undergone surgery, analysed in two different groups, over and under 65 years aged, with the purpose to determine whether preoperative variables and postoperative clinical outcome could differ between the two age groups.
The importance and novelty of this manuscript is due to the fact that the therapeutical management of meningiomas in elderly patients is still a debated question and the present study adds significant information to face up to this clinical issue. In fact, the originality of this extensive study consists in comparing clinical, radiological and surgical parameters with the clinical outcome among different age groups of meningioma patients.
The design and the experimental procedure of the study are well done, the presentation of the results is really rich of data, the methods are all correctly described, and both the discussion and the conclusions are consequent to the presented results.
Some minor revisions should be performed before publication:
- There is a problem in the result section with the figures since the sequence and the concordance between figures and their respective description in the text is often confusing and must be reviewed. For example, the resulting total number of figures is 9, but only 8 figures are described in the text. Figures 1 and 4 lack their own reference in the text. Figure 2 is totally lacking and there is only its figure legend.
- The English must be slightly revised throughout the text because there are some errors and few sentences need to be partially rewritten. For example, in the abstract, KSP must be written in the extended form in line 25.
- The authors should better control bibliography, since the sequence of references needs to be reordered inside the text because it isn’t correct; furthermore, there are some few errors to be corrected into the reference section.
Author Response
Response to reviewers: Thank you for your review and constructive suggestions to our work. As for the figures and tables, they are shown in order of appearance in the dedicated section of the template proposed by MDPI. Figure 1 can be found in the Material and methods section (we have highlighted it), Figure 9 has instead been included in the discussion as an example file of an elderly patient treated in our center. A review of grammar and vocabulary throughout the text was performed. The bibliography is now ordered according to appearance in the text.
Reviewer 2 Report
This is an interesting study with sound methodology and analysis. Since the prevalence of cardiopathy is higher in > 65 year age group, it may be adding to the worse outcomes. I would recommend doing a sub-group analysis if possible, to assess if the outcomes are different within this age group in patients without cardiopathy versus those with cardiopathy.
Author Response
Response to reviewers: Thanks for the interesting suggestion. It is absolutely feasible to perform a subgroup analysis in which cardiac subjects are not considered in the two groups, the main result being that unlike the main investigation we do not obtain a significant difference in the incidence of postoperative complications (p=0.128), while with regard to a stable reduction in post-operative KPS remains indifferent in the achievement of the result, confirming the hypothesis of an intrinsic frailty of the elderly subject to the recovery of its initial performance status.
However, we decided not to include it in the paper because of a lack of comparability between the groups obtained, in fact removing 79 cardiac patients in the Elderly group and only 29 in the Young group we obtain a difference between the two samples not statistically valid in the comparison. In the multivariate survey carried out instead, the variable "cardiopathy" was included in the contingencies, and in this way we reached the conclusions that can be deduced from the manuscript. In any case, following your suggestion, we have included this information in the results section of the KPS analysis.